# Detection of COVID-19 features in lung ultrasound images using deep neural networks
Lingyi Zhao [1], Tiffany Clair Fong [2] & Muyinatu A. Lediju Bell [1,3,4] ✉

## Abstract

**Background** Deep neural networks (DNNs) to detect COVID-19 features in lung ultrasound B-mode images have primarily relied on either in vivo or simulated images as training data. However, in vivo images suffer from limited access to required manual labeling of thousands of training image examples, and simulated images can suffer from poor generalizability to in vivo images due to domain differences. We address these limitations and identify the best training strategy.

**Methods** We investigated in vivo COVID-19 feature detection with DNNs trained on our carefully simulated datasets (40,000 images), publicly available in vivo datasets (174 images), in vivo datasets curated by our team (958 images), and a combination of simulated and internal or external in vivo datasets. Seven DNN training strategies were tested on in vivo B-mode images from COVID-19 patients.

**Results** Here, we show that Dice similarity coefficients (DSCs) between ground truth and DNN predictions are maximized when simulated data are mixed with external in vivo data and tested on internal in vivo data (i.e., $0.482 \pm 0.211$), compared with using only simulated B-mode image training data (i.e., $0.464 \pm 0.230$) or only external in vivo B-mode training data (i.e., $0.407 \pm 0.177$). Additional maximization is achieved when a separate subset of the internal in vivo B-mode images are included in the training dataset, with the greatest maximization of DSC (and minimization of required training time, or epochs) obtained after mixing simulated data with internal and external in vivo data during training, then testing on the held-out subset of the internal in vivo dataset (i.e., $0.735 \pm 0.187$).

**Conclusions** DNNs trained with simulated and in vivo data are promising alternatives to training with only real or only simulated data when segmenting in vivo COVID-19 lung ultrasound features.

## Plain Language Summary

Computational tools are often used to aid detection of COVID-19 from lung ultrasound images. However, this type of detection method can be prone to misdiagnosis if the computational tool is not properly trained and validated to detect image features associated with COVID-19 positive lungs. Here, we devise and test seven different strategies that include real patient data and simulated patient data to train the computational tool on how to correctly diagnose image features with high accuracy. Simulated data were created with software that models ultrasound physics and acoustic wave propagation. We find that incorporating simulated data in the training process improves training efficiency and detection accuracy, indicating that a properly curated simulated dataset can be used when real patient data are limited.

Multiple groups have demonstrated the potential of deep learning to aid COVID-19 diagnosis based on lung ultrasound image features[1–4], including A-lines[5], B-lines[6], and subpleural consolidations[7]. Among these features, B-lines are the most commonly seen features in lung ultrasound images of COVID-19 patients[8], appearing as laser-like vertical lines extending from the pleural line to the edge of the screen[9].

While most of the previous deep learning models implemented to detect COVID-19 features in lung ultrasound B-mode images primarily relied on in vivo labeled B-mode images as the training data, these datasets are difficult to obtain, and manually annotating in vivo data can be time consuming. Unlike in vivo data, simulated data can be generated on demand, in large quantities, with known ground truths. Previous work demonstrated that a training set containing a mixture of simulated and in vivo B-mode images enabled deep neural networks (DNNs) to achieve better performance when segmenting in vivo bone surface features and vessels[10,11]. In addition, when trained only on simulated raw ultrasound

[1]Department of Electrical and Computer Engineering, Johns Hopkins University, Baltimore, MD, USA. [2]Department of Emergency Medicine, Johns Hopkins Medicine, Baltimore, MD, USA. [3]Department of Biomedical Engineering, Johns Hopkins University, Baltimore, MD, USA. [4]Department of Computer Science, Johns Hopkins University, Baltimore, MD, USA. ✉e-mail: mledijubell@jhu.edu

channel data, DNNs can detect cyst-like features in both phantom and in vivo B-mode images[12].

Recent work implemented simulation-trained DNNs to identify in vivo B-line features in lung ultrasound images from COVID-19 patients[13]. The simulation-trained network found more B-line features than a human observer, which is promising for training less experienced users and triaging the most problematic cases in an emergency setting. In addition, with data augmentation included during the training process, Dice similarity coefficients (DSCs) between ground truth and DNN predictions were maximized[14]. Despite this promise, an automated post-processing algorithm was applied to remove false positives above the pleural line[13], which poses a limitation to real-time deployment. In addition, although previous work demonstrated that a training set containing a mixture of simulated and in vivo B-mode images enabled DNNs to achieve better performance when segmenting in vivo bone surface features and vessels[10,11], it is unclear whether or not this approach will improve DNN performance when segmenting lung ultrasound imaging features.

Generally, multiple self-supervised learning strategies have been investigated to address the challenge of limited labeled datasets in medical imaging. One strategy involves the application of self-supervised pre-training[15–19]. These investigations indicated that by pretraining neural networks with unlabeled images with self-supervised learning strategies, the network performance could be improved in downstream tasks, including brain tumor segmentation in 3D MRI[18], multi-organ segmentation in CT scan[19], and fetal standard scan plane classification in ultrasound imaging[16]. However, self-supervised pre-training also has limitations, including the need for large amounts of unlabeled data and substantial computational resources.

Another approach to address limited label datasets is to use simulated B-mode images to train deep learning models. Multiple groups have investigated using simulated B-mode images to train deep neural networks for segmentation tasks. Nair et al.[12,20] first proposed the use of simulated raw channel data to train DNNs to output B-mode images and segmentation maps of cysts. Training and testing datasets were simulated using the open-source Field II[21] ultrasound simulation software. Simulations included singular, water-filled, anechoic cysts surrounded by tissue. The transducer was modeled after an Alpinion L3-8 linear array transducer. Plane wave imaging was implemented with a single isonification angle of 0°. Results demonstrate the feasibility of employing deep learning as an alternative to traditional ultrasound image formation and segmentation with simulated datasets. Similarly, Behboodi et al.[22], Bhatt et al.[23], and Seoni et al.[24] used simulated B-mode images to train DNNs and test the simulation-trained DNNs on tissue-mimicking phantom data[22,24] and in vivo data[23]. Simulated B-mode images were generated with Field II[21], consisting of various combinations of hyperechoic lesions[22,24], anechoic lesions[22–24], lines[23], and point targets[23].

These initial findings collectively demonstrate that simulated B-mode images can be considered an alternative training dataset when real datasets are unavailable[12,20,22–24]. Despite the variety of shapes and structures included (e.g., circles[20,22,23] ellipses[22], lines[23], point targets[23]), Behboodi et al.[22] and Seoni et al.[24] only tested the resulting simulation-trained networks on phantom data, while Nair et al.[12] and Bhatt et al.[23] demonstrated that simulation-trained DNNs successfully segment structures in both phantom data and in vivo B-mode images of cysts surrounded by breast tissue.

Rather than using only simulated datasets in the training process, Patel et al.[10] investigated how simulated ultrasound data can be combined with in vivo data to improve the accuracy of DNNs in segmenting bone structures. A combination of 3D Slicer, SlicerIGT, and PLUS were used in order to obtain simulated US images and corresponding segmentations[25]. Results demonstrate that a DNN trained on a mixture of large-scale simulated datasets and limited in vivo datasets outperformed a network trained only on in vivo datasets when comparing the average Euclidean distance values. However, the in vivo training datasets had the same distribution as the in vivo test datasets, potentially limiting real-world generalization of the conclusions.

Our group is the first to implement simulation-trained DNNs to identify in vivo A-line, B-line, and consolidation features in lung ultrasound images from COVID-19 patients[13,14], demonstrating that simulation-trained DNNs can detect COVID-19 features in publicly available B-mode images from COVID-19 patients. The work presented herein integrates a new in vivo dataset that was acquired at the Johns Hopkins Hospital, from December 2021 to May 2022, containing differences in parameter distributions (e.g., image depth, image resolution, transducer frequency) relative to existing publicly available datasets[26]. As far as we are aware, no prior work investigates DNN performance when segmenting B-line features in COVID-19 patients with multiple training strategies, under multiple domain shift examples. Our investigated domain shifts combine simulated datasets with in vivo datasets, whether or not in vivo datasets from similar distributions as the test distribution are available.

This paper extends our simulation-trained approach[13,14] by incorporating multiple possible training strategies to ultimately identify the most suitable strategy for patient data. First, we train networks to segment B-line features with multiple combinations of training and testing strategies. Network performance was then tested with B-mode lung ultrasound images from COVID-19 patients. Finally, we compare the test DSCs of each network and determine the most suitable approach with respect to prediction accuracy and the number of required training epochs. This approach has implications for COVID-19 detection and for monitoring patients with long COVID and post-COVID syndrome[27–29].

When compared with our initial simulation-trained DNNs[13], employing data augmentation improves DSC scores by 20%, 165%, and 39%, respectively, when detecting in vivo A-line, B-line, and consolidation features. Follow-up strategy investigations demonstrate that when datasets from the same distribution as the test dataset are unavailable in the training process, including simulated B-mode images improves test DSC scores by 14% (Strategy 1) to 18% (Strategy 3) compared with including in vivo B-mode images only (Strategy 2). On the other hand, when datasets from the same distribution as the test dataset are available in the training process, combining simulated B-mode images and/or in vivo B-mode images from a different distribution improves test DSC scores by 1.5% (Strategy 5) to 3.3% (Strategy 6) compared with including in vivo B-mode images from the same distribution as test dataset only (Strategy 4). However, Strategy 4 required more epochs to train, indicating that the combination with simulated data remains as the more optimal solution. Our findings are promising for the development of future deep neural networks with simulated datasets included in the training process, regardless of the availability of in vivo data from the same distribution as test data.

## Methods
### Simulated data
**Data augmentation investigations.** To investigate the influence of data augmentation, a total of 30,000 lung phantoms were simulated with MATLAB based on publicly available in vivo lung ultrasound B-mode images with A-line, B-line, and consolidation features (10,000 phantoms per feature)[26,30]. The positions and the echogenicity of the features were changed to increase the variability. The specific type of B-line feature simulated for this task was discrete B-lines. To create each phantom, scatterers were randomly distributed in pre-defined regions to create A-line, B-line, or consolidation features, as well as background features including skin layers, muscle layers, pleural lines, and ribs. The amplitude of each scatterer was randomly chosen from a pre-defined range for each feature. Locations of pre-defined regions for each feature in each phantom were also randomly chosen from pre-defined boundaries containing three constraints to ensure realistic feature locations based on our observation in publicly available in vivo lung images. First, A-line, B-line, and consolidation features were constrained to be below pleural lines. Second, each phantom had a bat sign[30], a characteristic appearance of the pleural line along with the adjacent ribs. Third, background features such as skin layers and muscle layers were constrained to be above either ribs

or pleural lines. These simulated A-line, B-line, and consolidation features were utilized in our associated conference proceeding[14].

**B-line detection strategy investigations.** To determine the most appropriate strategy for B-line detection with data augmentation, a total of 10,000 lung phantoms were simulated with MATLAB based on publicly available in vivo lung ultrasound B-mode images[26,30], using the same phantom-creation process and associated constraints described under the Data augmentation investigations section above. Of these 10,000 simulated phantoms, 8000 contained B-line features and 2000 did not contain B-line features. To enhance the variety and realistic appearance of B-line features relative to our previous work[13,14], we simulated three additional types of B-line features, resulting in four specific types of B-line features (2000 phantoms per type): (1) discrete B-line only, (2) discrete B-line with attenuating artifacts below the pleural line, (3) confluent B-line only, and (4) confluent B-line with attenuating artifacts below the pleural line. The positions and the echogenicity of these four features were varied to increase the data variability. For each B-line detection strategy involving a simulated dataset, 1000 phantoms were randomly selected from each type of B-line feature, together with 1000 randomly selected phantoms containing no B-line features, resulting in a total of 5000 phantoms for each strategy.

**Image formation parameters & open dataset access.** We simulated raw channel data with the MATLAB Ultrasound Toolbox[31] using the phantoms described above (under the Data augmentation investigations and B-line detection strategy investigations section headings). The simulated transducer was a convex probe with 192 elements, a field of view of 73°, and a center frequency of 4 MHz. The simulated imaging depth was 10 cm, and the sampling frequency was 60 MHz. The parameters of this simulated transducer were the same as the parameters of the transducer used to acquire in vivo data from COVID-19 patients who were examined at Johns Hopkins (more details under the In vivo data section below). The simulated raw ultrasound channel data were then processed with delay-and-sum beamforming, demodulation, envelope detection, and scan conversion to generate B-mode images with a dynamic range of 60 dB (i.e., a common dynamic range when displaying ultrasound images). To demonstrate the similarity between simulated and real features, Fig. 1 shows real in vivo images (left) available in ref. 30, alongside examples of simulated B-mode images (right). The simulated B-mode images and paired ground truth segmentations described herein are available at https://gitlab.com/pulselab/covid19[32].

## In vivo data

The in vivo B-mode images for training and/or testing were derived from two datasets. The first dataset is a public dataset including B-mode images from COVID-19 patients worldwide (available at: https://github.com/jannisborn/covid19_ultrasound[26]), which will be referred to as the POCUS dataset for brevity. This POCUS dataset is the largest publicly available lung ultrasound dataset (202 videos + 59 images), comprising samples of COVID-19 patients, patients with bacterial pneumonia, (non-COVID-19) viral pneumonia, and healthy controls. We confirmed that this de-identified patient dataset follows best practices for ethics approval and consent. To test data augmentations, we included POCUS dataset B-mode images acquired with convex probes from COVID-19 patients with B-line and subpleural consolidation features, and from healthy controls with A-line features. In total, this POCUS test dataset included 32, 107, and 27 images originally marked as having A-line, B-line, and consolidation features by the POCUS dataset, respectively. Follow-up training for B-line detections with data augmentations using the POCUS dataset only included 100 B-mode images with B-line features and 15 B-mode images without B-line features of COVID-19 patients as the POCUS training dataset. As described by Born et al.[26], the POCUS dataset was sourced from community platforms, open medical repositories, health-tech companies, and other scientific literature (e.g., butterflynetwork.com, thepocusatlas.com, https://www.stemlynsblog.org, Northumbria Specialist Emergency Care

Hospital, theultrasoundjournal.springeropen.com, www.ncbi.nlm.nih.gov). Our labels for this dataset are publicly shared at https://gitlab.com/pulselab/covid19[32].

The second in vivo dataset contains B-mode images with B-line features acquired with a convex probe (Clarius C3HD) and Clarius Ultrasound App (v8.0.1, Clarius Mobile Health Corp.) from COVID-19 patients in the Emergency Department of Johns Hopkins Hospital, under Institutional Review Board Protocol IRB00310999 (which is a retrospective data analysis approval that does not allow public data sharing). Informed consent was not required because this retrospective study made secondary use of lung POCUS data collected as part of the standard clinical care of patients with suspected or confirmed COVID-19 infection. This dataset will be referred to as the JHH dataset for brevity. The JHH dataset included 958 B-mode images with B-line features. These images were obtained from 82 videos of 16 COVID-19 patients. All patients in the JHH dataset tested positive for COVID-19 through RT-PCR testing, which has a specificity of 99%[33]. To demonstrate the similarity between simulated and real features with the JHH dataset, Fig. 2 shows simulated B-mode images (left), alongside examples of in vivo images (right) from the JHH dataset with discrete B-lines and confluent B-lines.

## Training and testing dataset distributions

When investigating the influence of data augmentation, first the simulated data described under the Data augmentation investigations section were employed using an 80%-20% training-testing split. The resulting simulation-trained networks were then tested on the POCUS dataset B-mode images from healthy and COVID-19 patients described under the In vivo data section. Table 1 summarizes the details of this training and testing dataset with a null Strategy ID.

When exploring multiple training strategies for B-line detection with data augmentation, we investigated seven combinations of training and testing strategies that can be divided into two groups, using the data described under the B-line detection strategy investigations and the In vivo data section headings. First, for Strategies 1 to 3, the networks were trained only on the simulated dataset and/or POCUS dataset, then tested on the JHH dataset (i.e., the training dataset and the test dataset were drawn from different data distributions). Second, for Strategies 4–7, the networks were trained on 2/3 of JHH dataset, with or without various combinations of the POCUS and/or simulated datasets, then tested on the 1/3 of the held-out JHH dataset, which was unseen by the networks during the training process (i.e., the training and testing datasets contained the overlapping data distributions inherently embedded in the JHH dataset). Table 1 summarizes the details of training datasets and test datasets in each strategy. To provide a more accurate estimate of model generalization when incorporating the limited in vivo dataset in the training process (i.e., Strategies 2 to 7), we employed three-fold cross-validation.

To implement three-fold cross-validation for Strategies 2 and 3, the POCUS dataset was randomly split into three approximately equal-sized subsets on a video level (i.e., no images from the same video were present in multiple subsets). For each strategy, the networks were then trained on every possible combination of two of these three subsets, with or without the simulated dataset, resulting in three trained networks per strategy. For each test image in the JHH dataset, we first derived a probability map averaged across segmentation predictions from the three trained networks. We then computed the DSC between the probability map described above and the ground truth for each test image. To obtain the individual results for Strategies 2 and 3, the mean test DSC for the JHH dataset was obtained across the associated test DSCs for each test image and for each training epoch investigated.

To implement three-fold cross-validation for Strategies 4–7, the JHH dataset was randomly split into three approximately equal-sized subsets on a video level (i.e., to ensure no overlap of images from the same video in more than one subset). For each strategy, networks were trained on every possible combination of two of these three subsets and tested on the held-out subset, with or without various combinations of the POCUS and/or simulated

## In vivo and simulated examples

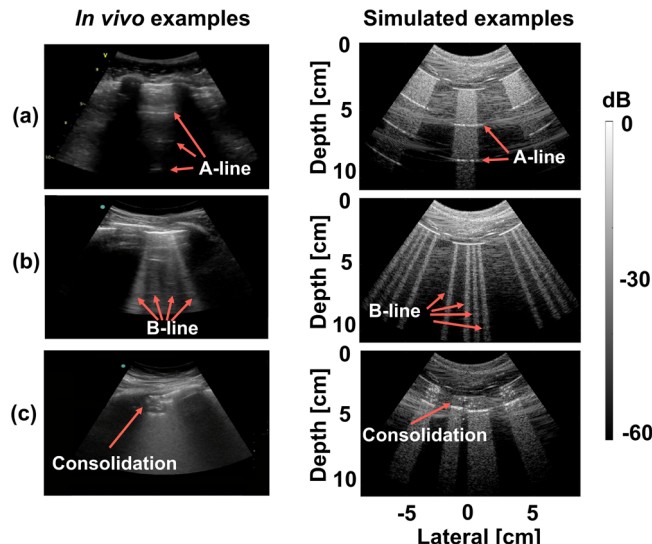

**Fig. 1 | In vivo and simulated examples.** In vivo (adapted from ref. 30 with permission from Elsevier) and simulated examples of (**a**) A-line, (**b**) B-line, and (**c**) consolidation features.

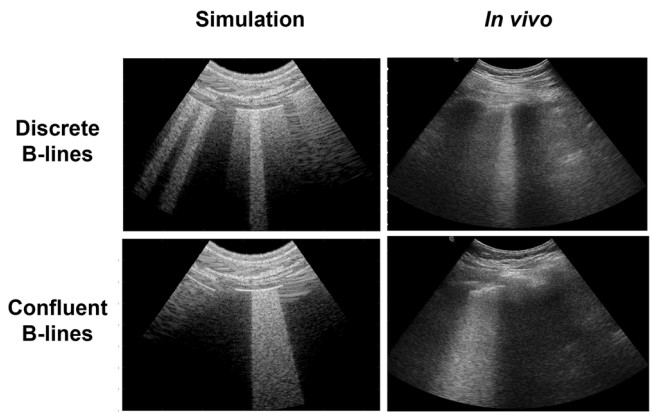

**Fig. 2 | Simulated and in vivo examples acquired at the Johns Hopkins Hospital.** Example simulated and in vivo B-mode images from the JHH dataset with discrete B-lines and confluent B-lines.

datasets, resulting in three trained networks per strategy and 958 total test images per strategy. To obtain the individual results for Strategies 4 through 7, the mean test DSC of the JHH dataset was obtained across the associated test DSCs for each test image in each held-out subset, as a function of each training epoch investigated.

These rigorous three-fold cross-validation approaches allowed us to average performance over different splits of the data, resulting in a more reliable performance estimation for the limited in vivo datasets. Due to the larger size of the simulated training datasets, we did not employ cross-validation for Strategy 1 and for the data augmentation investigations.

### Network architecture, data augmentation, and performance metrics

Despite the multiple training datasets, each network architecture was the same (i.e., universal for all datasets), though the training weights for each architecture differed. This universal architecture was based on the U-Net[34] architecture and a modified version of a previously reported deep learning architecture[12]. Specifically, instead of employing two separate decoders for segmentation and reconstruction, a single decoder was used for the segmentation task. Figure 3 shows the modified network architecture.

**Table 1 | Training and testing strategies for B-line detection with data augmentation, including the data augmentation investigations with a null Strategy ID**

| Strategy ID | Training dataset | Test dataset |
|---|---|---|
| null | Simulated | POCUS |
| 1 | Simulated | JHH |
| 2 | POCUS | JHH |
| 3 | Simulated, POCUS | JHH |
| 4 | 2/3 JHH | 1/3 JHH |
| 5 | Simulated, 2/3 JHH | 1/3 JHH |
| 6 | POCUS, 2/3 JHH | 1/3 JHH |
| 7 | Simulated, POCUS, and 2/3 JHH | 1/3 JHH |

*POCUS* Point of care ultrasound, *JHH* Johns Hopkins Hospital.

To detect features of interest, each DNN was trained for 80 epochs using the Adam optimizer[35], with a learning rate of 1e−5 and a mini-batch size of 16. The training loss was the DSC loss, which is defined as:

$$\text{DSCLoss}(\theta) = \frac{1}{n}\sum_{i=1}^{n}\left(1 - 2\frac{|S_{p,i}(I_d;\theta) \cap S_{t,i}|}{|S_{p,i}(I_d;\theta) + S_{t,i}|}\right) \quad (1)$$

where $S_{p,i}$ and $S_{t,i}$ are the vectorized segmentation masks for each training example, and $n$ is the total number of training examples in each mini-batch (i.e., the mini-batch size). Data were augmented by including horizontal flipping with a 0.5 probability, cropping and resizing with a predefined region, contrast adjustment, and Gaussian blur with a kernel size ranging from 3 to 25 and with a 0.8 probability. Our training was performed on a Tesla P40 GPU, and our code was developed using the PyTorch framework with Python 3.8. The average training time for each epoch was approximately 578 seconds.

### Statistical analyses

When comparing the impact of data augmentation, the mean ± standard deviation of test DSC scores achieved on the held-out test sets were measured per network per feature using the corresponding POCUS dataset described under the In vivo data section. When comparing B-line detections for Strategies 1 to 3, the performance of each network was measured with the mean ± standard deviation of test DSC scores of the entire JHH dataset, as indicated in Table 1. When comparing B-line detections for Strategies 4–7, each network performance was measured with the mean ± standard deviation of test DSC scores of the held-out JHH dataset, as indicated in Table 1. We tested each network performance on the test dataset and obtained a test DSC result after each training epoch. This process allows us to obtain a test DSC curve along different training epochs.

A Friedman test (two-sided) was implemented to determine statistically significant differences in the maximum DSC scores achieved for each strategy with a degree of freedom of 2 for Strategies 1-3 and 3 for Strategies 4-7. This test method was employed because test DSC scores were not normally distributed as indicated by the Kolmogorov–Smirnov test results. Significance was set at $p < 0.05$. We additionally quantified the presence of false positive and false negative B-line detections for each strategy investigated, as there is a potential for the presence of "false positive B-lines" (i.e., features that look like B-lines but are not actually B-lines[36]) or for B-lines to be missed by the network (i.e., false negative). To define a true positive, each connected component in the predicted segmentation map was considered as an individual B-line only if it satisfied the following criteria: (1) it contains at least 1% of the total pixels in a test image, (2) its orientation relative to horizontal direction exceeds 40 degrees (because B-lines typically have a vertical appearance rather than a horizontal appearance[5]), and (3) its eccentricity exceeds 0.8 (i.e., the major axis of the detected shape is longer

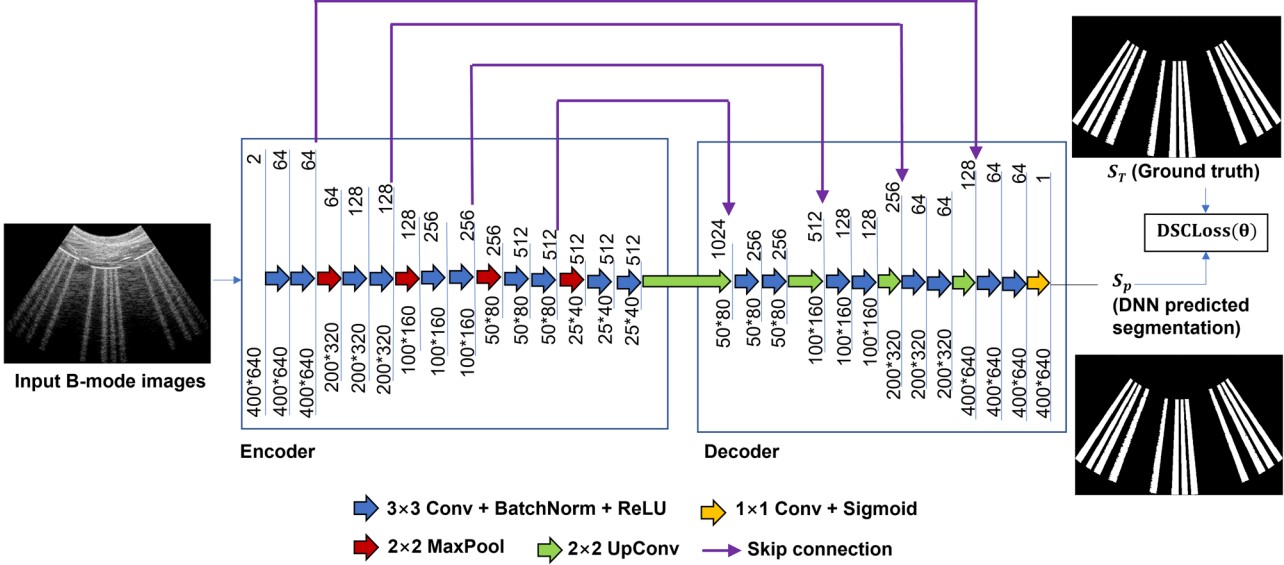

**Fig. 3 | DNN architecture implemented to segment features of interest with example input and output data.** U-Net architecture for DNN segmentation with example input B-mode image, DNN predicted segmentation, and ground truth.

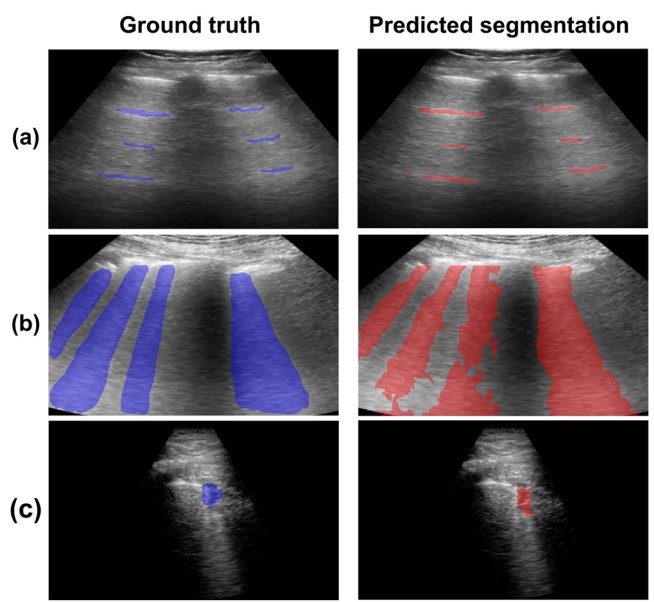

**Fig. 4 | Ground truth and predicted segmentations of multiple lung ultrasound features.** Ground truth (blue) and predicted segmentations (red) overlaid on lung ultrasound B-mode images from the POCUS dataset including healthy volunteers and COVID-19 patients: (**a**) A-line (healthy), (**b**) B-line (COVID-19), and (**c**) consolidation (COVID-19) features.

## Results
### Data augmentation benefits
Figure 4 shows example ground truth (blue) and predicted (red) segmentations overlaid on the corresponding in vivo POCUS B-mode images of each tested feature. These example results were selected from training epochs that achieved the highest averaged test DSCs. These images demonstrate that the predicted segmentation qualitatively achieves acceptable agreement with the ground truth. As summarized in Table 1, the null training Strategy ID implemented to obtain these results was most similar to that of Strategy 1 (i.e., only simulated datasets were used for training networks, although the specific datasets utilized differed).

Figure 5 shows the mean ± standard deviation DSC as a function of training epoch achieved with the POCUS in vivo test data when segmenting A-line, B-line, and consolidation features. In each case, the highest mean DSC was generally improved by employing data augmentation. Without data augmentation, the highest mean ± standard deviation DSCs were 0.40 ± 0.29 (epoch 1), 0.17 ± 0.15 (epoch 2), and 0.33 ± 0.35 (epoch 1) for A-line, B-line, and consolidation features, respectively. Employing data augmentation increased the highest mean ± standard deviation DSCs to 0.48 ± 0.29 (epoch 1), 0.45 ± 0.25 (epoch 20), and 0.46 ± 0.35 (epoch 75), representing 20%, 165%, and 39% improvement, respectively, when detecting in vivo A-line, B-line, and consolidation features. Based on these results, our data augmentation method appears to be most influential with respect to B-line segmentations, which supports our focus on applying this method to this feature in our training and testing strategy investigations in the following results.

### Different training & testing dataset distributions
Figure 6 shows three examples of ground truth B-line segmentations (overlaid in blue on the JHH B-mode images) and corresponding predictions obtained with Strategies 1, 2, and 3 (overlaid in red on the JHH B-mode images). In Fig. 6a, the example prediction with Strategy 1 had the fewest false positives, whereas that predicted with Strategy 2 had the most false positives. In Fig. 6b, Strategies 1, 2, and 3 each generated segmentation maps similar to the ground truth, with the segmentation map generated from Strategy 1 being closest to ground truth. In Fig. 6c, Strategy 1 generated a segmentation map with the most false negatives and Strategy 3 was the closest to ground truth. In addition, despite the false negatives in Strategy 1, the three strategies each successfully detected the presence of two B-lines, as indicated in the ground truth.

than its minor axis, which is characteristic of B-lines). We consider an individual B-line to be successfully detected when the DSC between the B-line, as predicted by DNNs, and the ground truth is larger than 0.4. These analyses were performed using MATLAB R2021a software (Natick, MA, USA). Because a clinically meaningful true negative definition does not exist in the context of the true positive and false positive described herein, specificity and sensitivity were not reported.

### Reporting summary
Further information on research design is available in the Nature Portfolio Reporting Summary linked to this article.

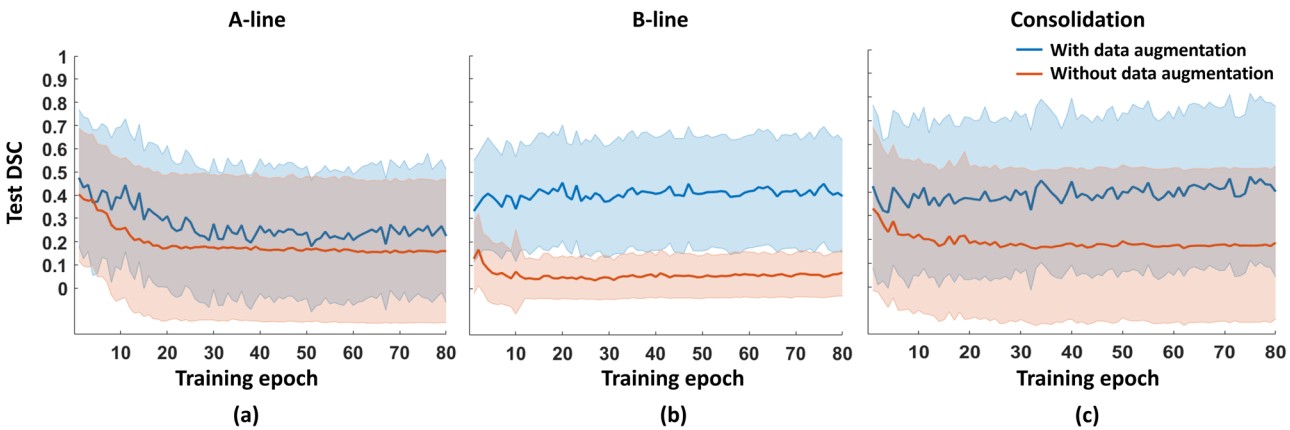

**Fig. 5 | Test DSC results obtained for multiple lung ultrasound features.** Mean test Dice similarity coefficient (DSC) per training epoch ± one standard deviation shown as shaded error bars (i.e., colored bands) when segmenting in vivo (**a**) A-line (*n* = 32 images), (**b**) B-line (*n* = 107 images), and (**c**) consolidation features (*n* = 27 images).

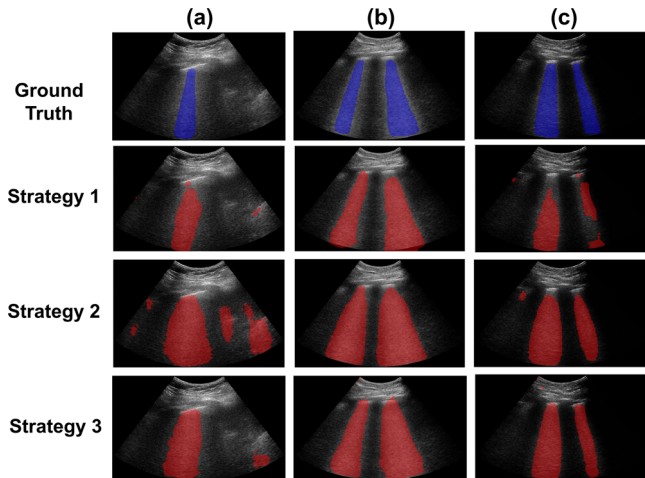

**Fig. 6 | Ground truth and predicted segmentations obtained with Strategies 1 to 3. a–c** Three examples of ground truth (blue) and predicted (red) B-line segmentations obtained with Strategies 1 to 3.

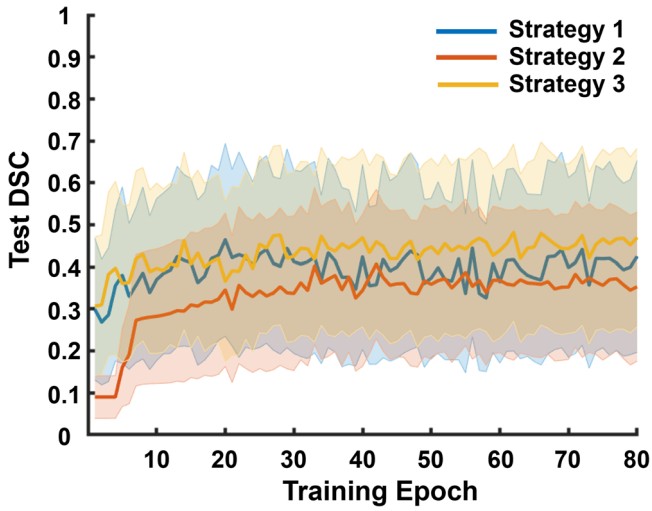

**Fig. 7 | DSCs obtained with Strategies 1 to 3.** Mean test Dice similarity coefficient (DSC) per training epoch ± one standard deviation shown as shaded error bars (i.e., colored bands) when segmenting B-line features with Strategies 1 to 3 (*n* = 958 images per strategy).

Figure 7 shows the mean ± standard deviation of test DSCs achieved with the in vivo test data as a function of training epoch. Employing a combination of the simulated and POCUS datasets during training (i.e., Strategy 3) most consistently achieved the highest test DSC. When training with only simulated B-mode images (i.e., Strategy 1), which was the same general approach employed in our previous conference paper[14], the performance was either approximately similar for early epochs or slightly worse for later epochs when compared to that of Strategy 3 (the difference between the maximum test DSC is less than 0.02). The worst performance was most consistently achieved when training with only the POCUS dataset (i.e., Strategy 2), which likely has a different data distribution than the test JHH dataset. Figure 7 also reports that the training epoch necessary to achieve the highest DSC in each training strategy varies. With the simulated training dataset (i.e., Strategy 1), the highest test DSC was achieved with the fewest training epochs (i.e., 20). With the simulated and POCUS training dataset (i.e., Strategy 3), the highest test DSC was achieved with the largest training epoch (i.e., 62). The highest test DSC for Strategy 2 was achieved at an epoch of 42, which is greater than that obtained with Strategy 1 but less than that obtained with Strategy 3.

The first three rows of Table 2 summarize the qualitative and quantitative results noted above. This table reports the maximum mean ± one standard deviation test DSCs among all the training epochs for each strategy, along with the training epoch at which the maximum mean test DSC was achieved. The differences between the mean DSCs reported in Table 2 for Strategies 1-3 were determined to be statistically significant ($p = 3.2 \times 10^{-93}$, $\chi^2 = 426$). This statistical significance was achieved, even though a total of 43%, 58%, and 61% of the test images in Strategies 1, 2, and 3, respectively, had false positive B-lines, while the corresponding percentage of false negative B-lines was 34%, 51%, and 18%, respectively.

### Overlapping training & testing dataset distributions

Figure 8 shows three examples of ground truth segmentations (blue) and corresponding predictions (red) obtained with Strategies 4–7. In Fig. 8a, segmentation maps generated with Strategies 5, 6, and 7 were similar to the ground truth, whereas that predicted with Strategy 4 had the most false positives. In Fig. 8b, Strategy 5 generated a segmentation map closest to ground truth, whereas that predicted with Strategy 4 had the most false positives. In Fig. 8c, the four strategies generated segmentation maps similar to ground truth, with Strategy 7 producing a segmentation map that was closest to ground truth. In addition, the four strategies each successfully detected the presence of one B-line, which was consistent with the ground truth.

Figure 9 shows the mean ± standard deviation of test DSCs achieved with the in vivo data as a function of training epoch. The combined simulated, POCUS, and JHH training datasets (i.e., Strategy 7) most consistently

**Table 2 | Performance of investigated training and testing strategies**

| Strategy ID | Training dataset | Test dataset | Maximum test DSC | Achieved at epoch |
|---|---|---|---|---|
| 1 | Simulated | JHH | 0.464 ± 0.230 | 20 |
| 2 | POCUS | JHH | 0.407 ± 0.177 | 42 |
| 3 | Simulated, POCUS | JHH | 0.482 ± 0.211 | 62 |
| 4 | 2/3 JHH | 1/3 JHH | 0.717 ± 0.185 | 72 |
| 5 | Simulated, 2/3 JHH | 1/3 JHH | 0.728 ± 0.186 | 57 |
| 6 | POCUS, 2/3 JHH | 1/3 JHH | 0.741 ± 0.185 | 72 |
| 7 | Simulated, POCUS, and 2/3 JHH | 1/3 JHH | 0.735 ± 0.187 | 66 |

*DSC* Dice similarity coefficient, *JHH* Johns Hopkins Hospital, *POCUS* point of care ultrasound.

achieved the highest test DSC, particularly at the earlier epochs. When training with only the JHH dataset (i.e., Strategy 4), the performance was the worst for early epochs or similar for later epochs when compared to that of Strategy 5, 6, or 7. In addition, Strategy 5 (i.e., combined simulated and JHH training datasets) achieved better performance for early epochs when compared to Strategy 6 (i.e., combined POCUS and JHH training datasets). Figure 9 also reports that the training epoch necessary to achieve the highest DSC in each training strategy varies. The combined simulated and JHH training dataset (i.e., Strategy 5) achieved the highest test DSC with the fewest training epochs (i.e., 57). The JHH training dataset and the combined JHH and POCUS training datasets (i.e., Strategies 4 and 6, respectively), achieved the highest test DSC with the largest training epoch (i.e., 72). With the combined simulated, POCUS, and JHH training datasets (i.e., Strategy 7), the epoch achieving the highest DSC was greater than that of Strategy 4 but less than that of Strategies 4 and 6 (i.e., 66).

The last four rows of Table 2 summarize these strategy results when there is an overlap of dataset distributions among the training and testing data. The maximum mean ± one standard deviation test DSCs among the 80 training epochs for each strategy are also reported. This approach generally achieved better performance than Strategies 1-3, which utilized completely separate training data distributions from the JHH test data. The differences between the mean DSCs reported in Table 2 for Strategies 4–7 were determined to be statistically significant ($p = 2.2 \times 10^{-32}$, $\chi^2 = 150$), with the exception of the difference between Strategies 6 and 7 ($p = 0.0565$, $\chi^2 = 3.6$). This statistical significance was achieved with false positive B-line detections in 14%, 12%, 12%, and 10% of the test images in Strategies 4, 5, 6, and 7, respectively, while the corresponding percentage of false negative B-lines was 11%, 11%, 10%, and 12%, respectively.

## Discussion

This paper is the first to investigate multiple possible training strategies for automated DNN segmentation of in vivo features in lung ultrasound images. Multiple possible training strategies previously included purely simulated or only in vivo data[1,13,14]. We now have a better understanding of appropriate combinations of these two classes of training strategies. When compared with our initial simulation-trained DNNs[13], employing data augmentation improved DSC scores by 20%, 165%, and 39%, respectively, when detecting in vivo A-line, B-line, and consolidation features. Without data augmentation, testing performance decreased after 1 or 2 epochs for A-line, B-line, and consolidation features (Fig. 5), suggesting substantial differences between the training and testing set. However, after applying data augmentation, an increase in test DSC in later epochs was observed for B-line and consolidation features (Fig. 5), suggesting that the previously noted differences between the training and testing set were mitigated. In addition, the proposed simulation-trained approach is not limited to the initially demonstrated B-line detections, as acceptable performance was also

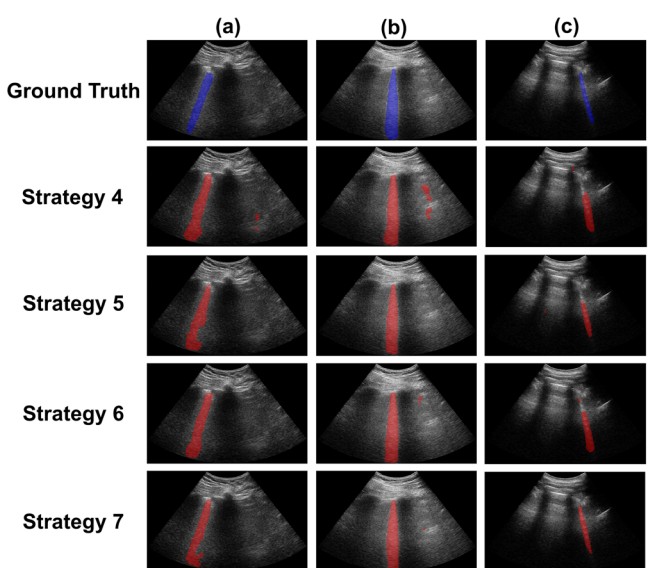

**Fig. 8 | Ground truth and predicted segmentations obtained with Strategies 4–7.** **a–c** Three examples of ground truth (blue) and predicted (red) segmentations obtained with Strategies 4–7.

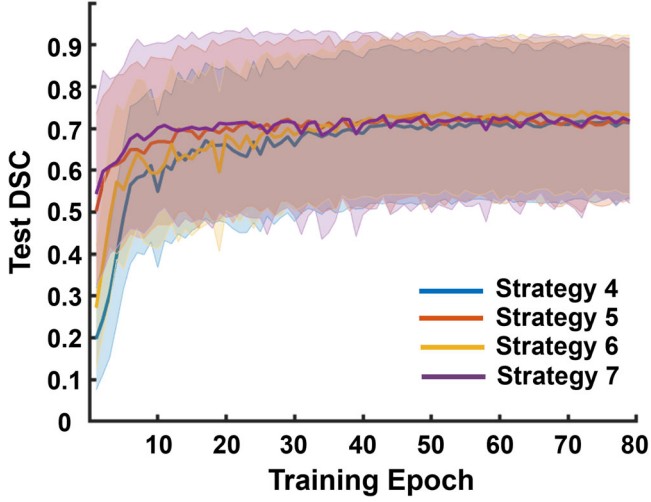

**Fig. 9 | DSCs obtained with Strategies 4–7.** Mean test DSC per training epoch ± one standard deviation shown as shaded error bars (i.e., colored bands) when segmenting B-line features with Strategies 4–7 (*n* = 958 images per strategy).

obtained for A-line and subpleural consolidation features. However, the greatest benefits of data augmentation were achieved for B-line detections (Fig. 5), which are therefore the primary focus of our follow-up strategy investigations, yielding the following four insights.

First, when datasets from the same distribution as the test dataset (i.e., JHH dataset) are available in the training process (Strategies 4–7), including simulated data in the training process (Strategies 5 and 7) consistently improved test DSC scores (compared to Strategy 4, which did not combine data). In addition, this approach reduced the necessary training epochs to achieve the highest test DSC, as shown in Fig. 9 and Table 2. While an even higher DSC was achieved by combining two types of in vivo data (i.e., Strategy 6), it took a greater number of epochs to reach this achievement. This group of approaches (i.e., overlapping training and testing dataset distributions) produced more favorable results than using completely different dataset distributions, as expected[10].

Second, when datasets from the same distribution are limited or unavailable (i.e., Strategies 1–3), combining simulated data with in vivo data

from a different distribution (i.e., Strategy 3) appears to be most effective based on Fig. 7 and Table 2. In addition, including only simulated B-mode images (i.e., Strategy 1) in the training process appears to be more effective than including only in vivo B-mode images (i.e., Strategy 2). For example, Strategy 1 improved test DSC scores by 14% when compared with Strategy 2 (i.e., $0.464 \pm 0.230$ vs $0.407 \pm 0.177$, $p = 2.6 \times 10^{-39}$). In terms of training epoch, 52% lower values were necessary for Strategy 1 to achieve its best performance when compared to that of Strategy 2 (i.e., 20 vs 42). Therefore, Strategy 1 may lead to a more efficient and cost-effective training approach. In addition to higher test DSC scores and fewer required training epochs to achieve the highest test DSC, another advantage of including simulated B-mode images as training data is the absence of required manual labeling (unlike in vivo B-mode images), which is tedious and time-consuming.

Third, in both classes of approaches described above, simulated datasets were combined with in vivo data from a different distribution than the test dataset (i.e., Strategies 3 and 7). This combination appears to improve test DSC scores when compared with including only the simulated dataset (i.e., Strategies 1 and 5) in the training process whether or not in vivo datasets from the same distribution as the test dataset are available for training. For example, when the JHH dataset was not present in the training process, combining the simulated dataset with the POCUS dataset (i.e., Strategy 3) improved test DSC score by 4% when compared to results obtained with the simulation-only training used in Strategy 1 (i.e., $0.482 \pm 0.211$ vs $0.464 \pm 0.230$, $p = 3.3 \times 10^{-7}$), as shown in Fig. 7 and Table 2. Similarly, when the JHH dataset was available in the training process, combining the simulated dataset with the POCUS dataset (i.e., Strategy 7) improved test DSC sore by 1% when compared to only combining with simulated data as in Strategy 5 (i.e., $0.735 \pm 0.187$ vs $0.728 \pm 0.186$, $p = 0.0049$) based on Fig. 9 and Table 2. It is worth noting that the network performance achieved when combining the in vivo dataset with the simulated dataset (Strategies 3 and 7) may be limited by the amount of available in vivo data. Therefore, we suggest a training approach that incorporates simulation data whenever possible, balanced by considerations of the limited availability of in vivo data. The extent to which the incorporation of simulated data depends on the availability of in vivo data remains to be determined. This open question is a possible area of investigation if in vivo data availability and access to associated hand-labeled annotations expand in the future.

Finally, when considering the seven strategies investigated, using only in vivo data drawn from a different distribution than the test data (i.e., Strategy 2) appears to present limitations with regard to test DSC scores or required training epochs. This is particularly true when compared with the combination of simulated and in vivo data in the training process, whether or not the in vivo training data were drawn from the same distribution as the test data. For example, when the JHH dataset was unavailable in the training process, only using the POCUS dataset in the training process (i.e., Strategy 2) obtained 16% lower test DSCs when compared to Strategy 3, which combined simulated and POCUS training data (i.e., $0.407 \pm 0.177$ vs $0.482 \pm 0.211$, $p = 8.0 \times 10^{-94}$), as shown in Table 2 and Fig. 7. When the JHH dataset was available for training, combining with the POCUS dataset (i.e., Strategy 6) required 9% more epochs to achieve the highest test DSC scores and generally required more epochs to achieve similar DSC values (i.e., $0.741 \pm 0.185$ vs $0.735 \pm 0.187$, $p = 0.0565$), when compared to Strategy 7, which combined simulated, POCUS, and JHH training datasets, as shown in Fig. 9. Therefore, based on these results, completely different in vivo training and testing dataset distributions should be avoided, particularly when simulated data are available to be combined with the in vivo training data and both training efficiency and training accuracy are desired.

The presented strategy investigations and resulting recommendations and suggestions focus on B-line features in lung ultrasound B-mode images, with justification based on the results in Fig. 5, which show the most gains with data augmentation applied to B-line detections and piqued our interest for additional investigations of this particular feature. We acknowledge that our findings with respect to the four most common types of B-lines (i.e., Figs. 7 and 9) may not necessarily be applicable to other features, particularly abnormal features in lung ultrasound B-mode images. In addition, the current simulated dataset is not representative of all real-world clinical scenarios, and the DNN performance may be further improved by enhancing the diversity of simulated B-mode images and incorporating a wider range of imaging artifacts and noise, which will be the focus of future work.

Additional future work can consider implementing Generative Adversarial Networks (GANs), incorporating motion tracking alongside B-line feature detection, and integrating B-line feature detection with patient demographics and clinical information (e.g., confounding variables such as age[37]). GANs were previously trained to produce both B-mode images and segmentation maps of cysts with raw radiofrequency data as the input[38], but they can also suffer from overfitting to training data[39–41]. Motion tracking can facilitate the visual detection of B-lines, and detect the presence of lung sliding, which is the movement between the two pleural layers that occurs during respiration. The combination of B-line and lung sliding detection can aid in the diagnosis of pneumonia and pneumothorax (in addition to COVID-19)[42]. Integrating B-line feature detection with relevant patient data may result in more accurate and personalized diagnosis, leading to the development of more comprehensive networks to aid diagnosis and monitoring of lung diseases, and ultimately providing a comprehensive approach to lung disease assessment.

Our primary objectives were to evaluate the accuracy and training efficiency of detecting B-line features in COVID-19 patients with multiple deep learning strategies. Detecting B-line features is an important step for evaluating COVID-19 or other pneumonia. In addition, among COVID-19 patients, there is a potential for the presence of "false positive B-lines" (i.e., features that look like B-lines but are not actually B-lines)[36]. Therefore, the accuracy of our network was additionally evaluated based on its ability to detect real B-lines and differentiate false positive B-lines from real B-lines. While the presence or absence of B-lines may be most important from a clinical perspective, an accurate segmentation is expected to increase our accuracy to quickly and efficiently determine the presence or absence of B-lines. Our study successfully identifies the most appropriate strategies for B-line segmentation using various combinations of available training datasets. We consider these contributions to be major foundational steps toward the ultimate goal of developing an effective clinical decision model, recognizing that a comprehensive diagnosis of COVID-19 relies on multiple factors.

## Data availability

We provide public access to our simulated datasets, paired ground truth segmentations, and segmentation labels for the in vivo POCUS dataset used herein at https://gitlab.com/pulselab/covid19[32]. The POCUS dataset is available at https://github.com/jannisborn/covid19_ultrasound[26]. These datasets can be accessed by downloading them from the above two links[26,32]. Source data for Figures 5, 7, and 9 are accessible in Supplementary Data 1. All other data are available from the corresponding author upon reasonable request.

## Code availability

All code underlying this article can be accessed from https://gitlab.com/pulselab/covid19[32] (https://doi.org/10.5281/zenodo.10324042[43]). MATLAB (R2010a) was used to develop code to generate simulated datasets. Python (v3.8) was used to develop code for building, training, and testing DNNs.

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

## Acknowledgements
This work was supported by NIH Trailblazer Award Supplement R21EB025621-03S1.

## Author contributions
M.A.L.B. conceived and initiated the study, supervised progress, provided advice and guidance, assisted with data interpretation and analysis, discussed manuscript outlines and contents, and edited manuscript drafts. L.Z. developed the code, performed the analysis, and prepared the manuscript. T.C.F. acquired the JHH dataset, assisted with feature segmentation, discussed medical concepts with L.Z. and M.A.L.B., and provided clinical perspectives and input. All authors read and approved the final manuscript.

## Competing interests
The authors declare no competing interests.
