## [Peer Review File · Communications Medicine]

Reviewers' comments:

Reviewer #1 (Remarks to the Author):

Compared to chest CT for COVID-19 lung infection detection, lung ultrasound B-mode imaging has the advantage of wider availability and lack of radiation exposure. The presented study, in combination with a previous conference publication by the same group, proposed to utilize simulated training data to hence the COVID-19 lung lesion segmentation using lung ultrasound B-mode images. This innovation was targeted to mitigate the limited size of the in vivo data and manual annotation. In addition, the study evaluated seven different strategies for formulating the training dataset to identify an optimal strategy. The study considered three dataset: a simulated dataset, a publicly available dataset (POCUS), and an inhouse collected dataset (JHH), with the latter mainly used for validation.

The document is well-written and well-structured. The findings were discussed in-depth and thoroughly. The potential application is clinical relevant. However, as stated by the author as a limitation, there were substantial differences between the simulated data distribution comparing to the real in vivo data. Thus, the benefit of mixing the simulated data in the training set is not that convincing based-on the presented results, though being a promising direction.

Comments to the authors

Abstract

1. Another popular approach to address "the burden of manual labeling" is to utilize self-supervised pretraining using unlabeled data, which has already been demonstrated to enhance medical image segmentation, e.g., <https://arxiv.org/abs/2006.03829>, and <https://arxiv.org/abs/2111.14791>. I would suggest to have a discussion to compare with this approach.

Methods

2. "To provide a more accurate estimate of model generalization when incorporating the limited in vivo dataset in the training process (i.e., Strategies 2 to 7), we employed three-fold cross-validation" - It is unclear how the three-fold cross-validation was conducted, especially for the strategy 2 and 3.

3. Fig 5 - The test DSC decreased starting from the epoch 1 in most cases, indicating the failure of the model training process, e.g., due to substantial difference between the training and test set. I would suggest to clearly state this in the discussion.

4. The training configuration is not provided, e.g., the learning rate, epochs, software platform (e.g., pytorch and the version), hard ware (what GPU?), and the time profiling (how long time it took for each epoch on average).

5. Fig 4 - Which strategy was used?

Results

6. Table II - If we compare strategy 7 to strategy 6, the test DSC decreased after inclusion of the

simulated data in the training set. This contradicts the statement in the abstract and Discussion that the combined approach yielded the best results.

7. Table II - The difference in best DSC between the strategies are small relative to the size of standard deviation, so it is difficult to tell if the differences are real (e.g., between 4, 5, 6, 7) or results of random noise. I would suggest to provide P values of the comparisons, at least for those led to the primary conclusions.

Reviewer #2 (Remarks to the Author):

The paper presents a method for detection of COVID-19 features in lung ultrasound by leveraging simulated and real data. The paper is well written, the experimental setup is clear and easy to follow. There are however a few areas in which the paper could be improved:

1. Novelty: In the current form the paper is missing an element of novelty, especially when positioning it against the earlier conference version of the paper. There are many research articles and systems in the space of ultrasound image processing that are trained from simulated / real data. Also the proposed architecture and training procedure are not novel. This field should be captured in a "related work" section (which is missing!) and the paper should be properly positioned in this context, clearly highlighting what are the technical contributions and how does it advance the state-of-the-art.

2. From table II it seems that there is a distribution shift affecting the performance for strategies 1,2,3. As soon as JHH data is added to the training, the performance increases significantly in strategies 4,5,6,7. Are the differences between 4,5,6,7 significant? This increase observed with the addition of the JHH data in the training is a bit concerning; implying the the POCUS dataset is not by all means sufficient to close the performance gap. Ideally, the performance should be quantified on a 3rd source of real data to demonstrate the generalizability of the system.

Reviewer #3 (Remarks to the Author):

This work aims to investigate the potential use of synthetic ultrasound data for training deep neural networks to detect signs of COVID-19 on lung ultrasound. This is an important area of research, as medical AI is often hampered by limitations in the size of available labelled training data.

The work is well-written, and clearly laid out, but I have the following comments / questions:

- There is little description about how the synthetic data are made, other than saying 'simulated in MATLAB'. Could more information be given about this? I'm not sure I understand the relationship between the 'publicly available in vivo lung images' and the 'phantoms'.

- For the in vivo data, how were the patients classified? In what way were the patients diagnosed with COVID, and what is the chance of this being incorrect? Much has been written about the poor quality of available covid imaging datasets, with many including a high percentage of false positives.

- Who are the healthy controls, and why have they had a lung ultrasound? Are they comparable with

the disease group in terms of age, and other demographics? Again, much has been written about confounders such as age leading to incorrect conclusions (for example a model learning differences in healthy vs disease based on an irrelevant factor such as age).

- For most of the training/testing strategies, the test set consisted only of the JHH dataset- but this dataset seems to only have patients with COVID in it (i.e. no controls). Does this have implications for generalisability? Given that the point of lung ultrasound is to detect and diagnose lung disease such as covid, it seems unwise to test it only in a group with a confirmed diagnosis- for example if it gives lots of false positive B-lines in normal lungs, this would be unhelpful and potentially dangerous.

- Fig 7- the shaded area is not defined in the legend, I assume it is +/- 1 SD? This seems very wide, perhaps reflecting a small test set?

- From a clinical perspective, it is the presence or absence of B lines that is important, not how accurately one can segment them- the segmentation is not useful for the diagnosis. Could the test set be reanalysed on that basis, to define how clinically useful these models will be in making a covid diagnosis, and allowing calculation of sensitivity and specificity.

- Based on fig 7 and fig 9, my conclusion would be that if you include similar data in your training and test set (i.e. JHH data in both), performance increases. This does not seem that surprising, and I think is well described. From the figures it looks like the simulated data essentially makes no difference to test set performance.

- I think given that you are making multiple statements about different strategies giving difference results, it would be important to test these statements statistically. For example, saying that strategy 1 improved results by 14% seems incorrect, given that both strategies are both well within 1 SD of each other. Surely that 14% could be accounted for by random chance?

- Finally, in the era of foundation models, have you done any work on using generative AI to synthesise ultrasound images? These seems like it would be the most promising way of improving your results.

We are grateful to the editors for handling our manuscript and to the three reviewers for taking the time to read our manuscript and provide thoughtful and insightful critiques. We carefully considered each comment and used these reviews to make updates, resulting in what we believe is a significantly stronger manuscript. New text in the revised manuscript is indicated with red lettering, and our point-by-point response to reviewers' comments appears below.

Reviewer #1

(Expertise: MSc, deep learning, radiology, chest)

Compared to chest CT for COVID-19 lung infection detection, lung ultrasound B-mode imaging has the advantage of wider availability and lack of radiation exposure. The presented study, in combination with a previous conference publication by the same group, proposed to utilize simulated training data to hence the COVID-19 lung lesion segmentation using lung ultrasound B-mode images. This innovation was targeted to mitigate the limited size of the *in vivo* data and manual annotation. In addition, the study evaluated seven different strategies for formulating the training dataset to identify an optimal strategy. The study considered three dataset: a simulated dataset, a publicly available dataset (POCUS), and an inhouse collected dataset (JHH), with the latter, mainly used for validation.

The document is well-written and well-structured. The findings were discussed in-depth and thoroughly. The potential application is clinical relevant.

Thank you!

However, as stated by the author as a limitation, there were substantial differences between the simulated data distribution comparing to the real *in vivo* data. Thus, the benefit of mixing the simulated data in the training set is not that convincing based-on the presented results, though being a promising direction.

Thanks for appreciating the promising direction! To address this concern regarding the benefit of mixing datasets, it is important for us to first clarify our motivation for including a simulated dataset during training. While it is true that simulated data is different from *in vivo* data, it is also common for *in vivo* datasets available during training to originate from a different distribution than *in vivo* datasets during clinical implementation. This is worth noting because models trained and tested on data from the same distribution may not generalize well to unseen data from different distributions, patients, hospitals, etc.

Given the above consideration, our study aimed to investigate whether including a simulated dataset during training can enhance *in vivo* test performance when compared with using *in vivo* datasets alone, especially when the *in vivo* training dataset is from a different distribution than *in vivo* test dataset (e.g., Strategy 2 vs. Strategy 3). Our results indicated that when the *in vivo* training dataset came from a

different distribution than the *in vivo* test dataset, including the simulated dataset (Strategy 3) improved test DSC by 18.4% compared with using the *in vivo* dataset alone (Strategy 2) (i.e., 0.482 ± 0.211 vs. 0.407 ± 0.177 , $p < 0.05$). In addition, when the *in vivo* training dataset came from the same distribution as the *in vivo* test dataset, including the simulated dataset (Strategy 5) improved test DSC by 1.5% when compared with using the *in vivo* dataset alone (Strategy 4) (i.e., 0.728 ± 0.186 vs. 0.717 ± 0.185 , $p < 0.05$) and required 21% fewer epochs (57 vs. 72) . Therefore, we consider improvements in terms of both accuracy through higher DSCs and training efficiency through reduced epochs required. From this perspective, we conclude that our results are promising for future developments of deep neural networks with simulated datasets included in the training process.

Abstract

1. Another popular approach to address "the burden of manual labeling" is to utilize self-supervised pretraining using unlabeled data, which has already been demonstrated to enhance medical image segmentation, e.g., <https://arxiv.org/abs/2006.03829>, and <https://arxiv.org/abs/2111.14791>. I would suggest having a discussion to compare with this approach.

Thanks for the suggestion. We added a Related Work section (as suggested by Reviewer #2) and discussed the self-supervised pre-training approach in the first paragraph of this section (i.e., Section II-A).

2. "To provide a more accurate estimate of model generalization when incorporating the limited *in vivo* dataset in the training process (i.e., Strategies 2 to 7), we employed three-fold cross-validation" - It is unclear how the three-fold cross-validation was conducted, especially for the strategy 2 and 3.

We apologize for the lack of clarity. We addressed this concern by providing a more comprehensive description of our three-fold cross-validation approach in the third and fourth paragraphs of Section III-C.

3. Fig 5 - The test DSC decreased starting from epoch 1 in most cases, indicating the failure of the model training process, e.g., due to substantial difference between the training and test set. I would suggest to clearly state this in the discussion.

As suggested, we included this detail in the first paragraph of Section V.

4. The training configuration is not provided, e.g., the learning rate, epochs, software platform (e.g., PyTorch and the version), hardware (what GPU?), and the time profiling (how long time it took for each epoch on average).

We added these important training configurations to the second paragraph of Section III-D.

5. Fig 4 - Which strategy was used?

Only simulated datasets were used in the training process for the results displayed in Fig. 4. Although the training datasets were not the same as those used in Strategy 1, the training strategy was the same as Strategy 1 (i.e., only simulated datasets were used for training networks). This detail was added to the first paragraph of Section III-C, and it is now further clarified in the first paragraph of Section IV-A.

Results

6. Table II - If we compare strategy 7 to strategy 6, the test DSC decreased after inclusion of the simulated data in the training set. This contradicts the statement in the abstract and Discussion that the combined approach yields the best results.

We agree that the test DSC decreased by 0.8% (i.e., 0.741 ± 0.185 vs. 0.735 ± 0.187) after including the simulated dataset in the training process. However, this difference is not statistically significant ($p > 0.05$). In addition, we also observed that the epochs needed to achieve the highest test DSC decreased by 8% (66 vs. 72 epochs), suggesting higher training efficiency when including the simulated dataset. We described this observation in the second paragraph of Section V when stating, "While an even higher DSC was achieved by combining two types of *in vivo* data (i.e., Strategy 6), it took a greater number of epochs to reach this achievement." We additionally clarified in the Abstract and fifth paragraph of Section V that our statements were made with regard to both training accuracy and training efficiency.

7. Table II - The difference in best DSC between the strategies are small relative to the size of standard deviation, so it is difficult to tell if the differences are real (e.g., between 4, 5, 6, 7) or results of random noise. I would suggest to provide P values of the comparisons, at least for those led to the primary conclusions.

As suggested, we conducted a Friedman test to assess the statistical significance of these differences. The p-values confirm that the observed differences in DSC between these strategies are statistically significant, except for the comparison between Strategy 6 and Strategy 7.

Reviewer #2

(Expertise: MSc, deep learning, ultrasound, lung)

The paper presents a method for the detection of COVID-19 features in lung ultrasound by leveraging simulated and real data. The paper is well written, the experimental setup is clear and easy to follow.

Thank you!

There are however a few areas in which the paper could be improved:

1. Novelty: In the current form the paper is missing an element of novelty, especially when positioning it against the earlier conference version of the paper. There are many research articles and systems in the space of ultrasound image processing that are trained from simulated/real data. Also the proposed architecture and training procedure are not novel. This field should be captured in a "related work" section (which is missing) and the paper should be properly positioned in this context, clearly highlighting what are the technical contributions and how does it advance the state-of-the-art.

Thank you for the suggestion. As recommended, we added a Related Work Section (Section II) to the revised manuscript. Section II-B discusses existing work on simulation training, and Section II-C highlights our technical contributions and how they advance the state-of-the-art.

2. From table II it seems that there is a distribution shift affecting the performance for strategies 1,2,3. As soon as JHH data is added to the training, the performance increases significantly in strategies 4,5,6,7. Are the differences between 4,5,6,7 significant?

We conducted a Friedman test to evaluate the statistical significance of the differences between Strategies 4 through 7 and added the resulting details about the statistical significance to the revised manuscript. In particular, the p-values confirm that the observed differences in DSC between these strategies are statistically significant, except for the comparison between Strategy 6 and Strategy 7.

This increase observed with the addition of the JHH data in the training is a bit concerning; implying that the POCUS dataset is not by all means sufficient to close the performance gap. Ideally, the performance should be quantified on a 3rd source of real data to demonstrate the generalizability of the system.

We agree that the POCUS dataset is not sufficient to close the performance gap, largely due to its distribution variance from the JHH dataset. This disparity in training and testing datasets often mirrors real-world scenarios, motivating us to

investigate how *in vivo* training datasets — when differing from *in vivo* test datasets — impact the testing performance. Interestingly, we found that when the *in vivo* training dataset is from a different distribution than the *in vivo* test dataset, including a simulated dataset during training (Strategy 3) can enhance *in vivo* test performance when compared with using *in vivo* datasets alone (Strategy 2).

Regarding the request for a third data source, the acquisition of an additional source of real, high-quality, B-mode image data is challenging when considering privacy concerns with gaining access to data outside of our institution and the resourcefulness we already employed to utilize the largest publicly available COVID-19 lung ultrasound repository, as summarized in our recent review article [1]. In addition, note that it is not accurate to consider that we only have two real data sources (e.g., JHH and POCUS), as the POCUS dataset contains data from at least six different sources worldwide. We hope that this particular clarification (also added to Section III-B) addresses the concern that led to the request for a third (more accurately at least eighth) data source.

To address the concern regarding generalizability, we wish to clarify that we were careful to implement a three-fold cross-validation approach whenever *in vivo* datasets were included in the training process. Our three-fold cross-validation approach allowed us to average network performance over different data splits, resulting in a more reliable performance estimation for the limited *in vivo* dataset and effectively containing at least three real data sources for Strategies 4-7. (i.e., POCUS, 2/3 JHH exclusively for training, 1/3 JHH exclusively for testing). We added more details about our rigorous three-fold cross-validation methods to Section III-C.

Reviewer #3

(Expertise: MD, cardiovascular imaging, diagnosis)

This work aims to investigate the potential use of synthetic ultrasound data for training deep neural networks to detect signs of COVID-19 on lung ultrasound. This is an important area of research, as medical AI is often hampered by limitations in the size of available labeled training data.

The work is well-written, and clearly laid out, but I have the following comments/questions:

Thank you!

1. There is little description about how the synthetic data are made, other than saying 'simulated in MATLAB'. Could more information be given about this? I'm not sure I understand the relationship between the 'publicly available *in vivo* lung images' and the 'phantoms'.

As requested, we added more details about our phantom simulation process to Section III-A1. The phantoms can be considered as synthetic data modeled after real, physical and anatomical constraints, whereas the *in vivo* data is real data.

2. For the *in vivo* data, how were the patients classified? In what way were the patients diagnosed with COVID, and what is the chance of this being incorrect? Much has been written about the poor quality of available covid imaging datasets, with many including a high percentage of false positives.

For the *in vivo* dataset, patients were classified as either healthy or COVID-19 positive based on RT-PCR tests. As indicated in a prior study [29], the specificity of the RT-PCR test is 0.99 ± 0.01 , which corresponds to a false positive rate of 0.01. We recognize the concern about high false positive rates derived from imaging diagnoses. This understanding motivated us to rely on RT-PCR tests for patient classification, ensuring higher accuracy. We added these important details to the last paragraph of Section III-B.

3. Who are the healthy controls, and why have they had a lung ultrasound? Are they comparable with the disease group in terms of age and other demographics? Again, much has been written about confounders such as age leading to incorrect conclusions (for example a model learning differences in healthy vs disease based on an irrelevant factor such as age).

The B-mode images from the healthy group in our study were sourced from a public dataset (https://github.com/BorgwardtLab/covid19_ultrasound). As these images were compiled from various online sources, it is challenging to determine the comparability of the healthy subjects with the disease group in terms of age and other demographics. We included the healthy group in our study to assess the network's performance in detecting the A-line features commonly observed in healthy subjects. We did not incorporate healthy controls for detecting B-line and consolidation features as these features typically manifest in diseased lungs. Our primary objective is to evaluate the detection of A-line, B-line, and consolidation features using different deep learning strategies (as opposed to directly comparing healthy and diseased groups). Thanks for noting potential confounders such as age. We clarified this important concern as one possible direction for future research in the last paragraph of Section V. Unfortunately, we are unable to address this concern at this time with the existing data collected for the presented study, as we did not record age for all cases investigated.

4. For most of the training/testing strategies, the test set consisted only of the JHH dataset- but this dataset seems to only have patients with COVID in it (i.e. no controls). Does this have implications for generalisability? Given that the point of lung ultrasound is to detect and diagnose lung diseases such as covid, it seems

unwise to test it only in a group with a confirmed diagnosis- for example, if it gives lots of false positive B-lines in normal lungs, this would be unhelpful and potentially dangerous.

Our primary objective is to evaluate the accuracy of detecting B-line features in COVID-19 patients with multiple deep learning strategies because detecting B-line features is an important step for evaluating COVID-19 or other pneumonia. In addition, even among COVID-19 patients, there is a potential for the presence of “false positive B-lines” (i.e., features that look like B-lines but are actually not B-lines). The accuracy of our network was evaluated based on its ability to detect real B-lines and differentiate “false positive B-lines” from real B-lines. This detail was clarified in the second paragraph of Section VI.

5. Fig 7- the shaded area is not defined in the legend, I assume it is +/- 1 SD? This seems very wide, perhaps reflecting a small test set.

The shaded area represents +/- one standard deviation, as detailed in the figure caption of Fig. 7 and Fig. 9. The JHH test dataset included 958 B-mode images, which is not too small for a test set. The observation regarding standard deviations further underscores the importance of evaluating deep learning strategies with different combinations of datasets, particularly when the available training dataset does or does not share the same distribution as the test dataset.

6. From a clinical perspective, it is the presence or absence of B lines that is important, not how accurately one can segment them- the segmentation is not useful for the diagnosis. Could the test set be reanalysed on that basis, to define how clinically useful these models will be in making a covid diagnosis, and allowing calculation of sensitivity and specificity?

We re-evaluated our results and determined there were 43-61% false positive B-line detections for Strategies 1-3 and 10-14% false positive B-line detections for Strategies 4-7, as noted in the text added to Sections IV-B and IV-C. Hence, there is a correlation between these measurements and accurate segmentation (i.e., Strategies 4-7 consistently performed better than Strategies 1-3 across these multiple metrics). Given this correlation, our results support the notion that accurate segmentation is essential for the successful detection of B-line features. An accurate segmentation is expected to increase our accuracy (and the accuracy of automated algorithms) to quickly and efficiently determine the presence or absence of B-lines. In addition, our study primarily aimed to identify the most appropriate strategies for B-line segmentation using various combinations of available training datasets. We consider these contributions to be major foundational steps toward the ultimate goal of developing an effective clinical decision model, recognizing that a comprehensive diagnosis of COVID-19 relies on multiple factors. We summarized these details in the last paragraph of Section VI.

We were unable to calculate specificity because it is not possible to isolate “true negatives” in our case. Based on our definitions of true and false positives, a “true negative” would be defined as no B-line detected when there should be no B-line detected, resulting in true negative rate (or specificity) defined as:

Specificity = (number of true negative pixels)/(number of true negative pixels + false positive pixels)

However, this calculation is basically a calculation of overlapping negative pixels (similar to DSC) rather than a report of “true negative” B-lines. Therefore, the definition of true negative as everything outside of our segmentation mask is a meaningless metric to report from a clinical perspective. In addition, although diseases may have negative diagnoses, there are no true negative B-lines because all B-lines presented in ground truth are positive B-lines. As specificity is meaningless, sensitivity was also not reported because it is not fully meaningful from a statistical perspective without reporting specificity. This rationale is summarized in Section III.E.

7. Based on Fig 7 and Fig 9, my conclusion would be that if you include similar data in your training and test set (i.e. JHH data in both), performance increases. This does not seem that surprising, and I think is well described. From the figures, it looks like the simulated data essentially makes no difference to test set performance.

We agree that including datasets from the same distribution for both the training and testing processes increases the performance of deep neural networks (e.g., Strategies 4-7 vs. Strategies 1-3, as reported in Table II). However, our findings also indicate an important role for the simulated dataset. Specifically, when the JHH dataset was available in the training process, including the simulated dataset in the training process (Strategy 5) improved test DSC scores by 1.5% (0.728 ± 0.186 vs 0.717 ± 0.185 , $p < 0.05$) and required 21% fewer epochs (57 vs 72), when compared to using the JHH dataset alone (Strategy 4). These results indicate that the simulated dataset contributed positively to both the accuracy and training efficiency of the network.

8. I think given that you are making multiple statements about different strategies giving different results, it would be important to test these statements statistically. For example, saying that strategy 1 improved results by 14% seems incorrect, given that both strategies are both well within 1 SD of each other. Surely that 14% could be accounted for by random chance?

We first conducted the Kolmogorov-Smirnov test and found that the dependent variable, test DSC scores, was not normally distributed. We then conducted a

Friedman test to evaluate the statistical significance of the differences among Strategies 1 through 3 and among Strategies 4 through 7. We observed significant differences between these strategies ($p < 0.05$) except for between Strategy 6 and Strategy 7 ($p = 0.0565$). We added these details about statistical significance to Sections III-E and IV.

9. Finally, in the era of foundation models, have you done any work on using generative AI to synthesise ultrasound images? These seems like it would be the most promising way of improving your results.

The use of generative AI to synthesize ultrasound images is not well developed, and it is therefore inferior to models that are based on a physically realistic simulation process. At the same time, we acknowledge its potential, and our previous study [26] demonstrated that Generative Adversarial Network (GAN) can be successfully trained to produce both B-mode images and segmentation maps of the cyst with raw radiofrequency data as input. We added this possible future direction to the last paragraph of Section V.

REVIEWERS' COMMENTS:

Reviewer #1 (Remarks to the Author):

Thank you for considering my comments and addressing them comprehensively. I have no further comments for the authors.

Reviewer #2 (Remarks to the Author):

I would like to thank the authors for the thorough revision and for answering my questions in detail - particularly the related work section and statistical significance testing help the reader to get a greater insight and understanding.

Given your extensive prior work in this space, I am still concerned about the technical novelty of the paper (in comparison).

Reviewer #3 (Remarks to the Author):

The authors seem to have made careful and thoughtful responses to the reviewers suggestions and questions.

We are grateful to the editors for handling our manuscript and to the three reviewers for taking the time to read our manuscript again after we carefully considered each previous comment and made updates. As requested, our point-by-point response to reviewers' comments appears below.

Reviewer #1

(Expertise: MSc, deep learning, radiology, chest)

Thank you for considering my comments and addressing them comprehensively. I have no further comments for the authors.

Thanks so much for all of your comments!

Reviewer #2

(Expertise: MSc, deep learning, ultrasound, lung)

I would like to thank the authors for the thorough revision and for answering my questions in detail - particularly the related work section and statistical significance testing help the reader to get a greater insight and understanding.

We appreciate your gratitude with respect to these details.

Given your extensive prior work in this space, I am still concerned about the technical novelty of the paper (in comparison).

Our prior work in this space is limited to brief, page-limited conference papers that contain insufficient details to fully understand or replicate our contributions. We also confirm that we are in compliance with the journal's policy on conference papers (<https://www.nature.com/commsmed/editorial-policies/preprints-conference-proceedings>), which states:

"Publishing work in conference proceedings is common in some research communities. The Nature journals are happy to consider submissions containing material that has been published in a conference proceedings paper. However, the submission should provide a substantial extension of results, methodology, analysis, conclusions and/or implications over the conference proceedings paper; the final decision on what constitutes a substantial extension is made by the editors at each individual journal. Authors must provide details of the conference proceedings paper with their submission including relevant citation in the submitted manuscript. Authors must obtain all necessary permissions to re-use previously published material and attribute appropriately."

Thanks again for all of your feedback and insights.

Reviewer #3

(Expertise: MD, cardiovascular imaging, diagnosis)

The authors seem to have made careful and thoughtful responses to the reviewers' suggestions and questions.

Thanks so much for all of your questions and suggestions!